# Capsaicin for Weight Control: “Exercise in a Pill” (or Just Another Fad)?

**DOI:** 10.3390/ph15070851

**Published:** 2022-07-11

**Authors:** Arpad Szallasi

**Affiliations:** Department of Pathology and Experimental Cancer Research, Semmelweis University, 1085 Budapest, Hungary; Szallasi.Arpad@med.semmelweis-univ.hu

**Keywords:** capsaicin, CAPSIMAX, capsiate, TRPV1, gut microbiota, obesity, weight control

## Abstract

Medical management of obesity represents a large unmet clinical need. Animal experiments suggest a therapeutic potential for dietary capsaicin, the pungent ingredient in hot chili peppers, to lose weight. This is an attractive theory since capsaicin has been a culinary staple for thousands of years and is generally deemed safe when consumed in hedonically acceptable, restaurant-like doses. This review critically evaluates the available experimental and clinical evidence for and against capsaicin as a weight control agent and comes to the conclusion that capsaicin is not a magic “exercise in a pill”, although there is emerging evidence that it may help restore a healthy gut microbiota.

## 1. Introduction

With almost 40% of adults in the world being overweight or obese, obesity has reached pandemic proportions [1]. Indeed, the worldwide prevalence of obesity nearly tripled between 1975 and 2016 [1]. Even worse, childhood obesity is on the rise: already one in five children are overweight. Most authorities agree that lifestyle changes (less exercise and increased access to unhealthy fast food) are primarily to blame for this trend.

Type-2 diabetes (T2DM) is of particular concern in countries with a high prevalence of obesity. For example, the US (the country with the 12th highest obesity rate in the world at 36%) already has 37 million T2DM patients with an estimated annual cost of USD327 billion in medical care and lost productivity [2].

Maintaining normal body weight is essential in preventing common diseases such as hypertension, hyperinsulinemia, insulin resistance, and T2DM, pathologies that comprise the metabolic syndrome. Indeed, losing weight is the first step that is recommended by doctors for overweight people with increased blood pressure. Unfortunately, it is easier said than done. Many patients are unable to lose weight solely by changing diet and increasing physical activity. Surgical means of weight loss (for example, bariatric surgery, gastric banding, or stapling) may have serious side-effects [3]. Clearly, there is a pressing need for drugs that can help people to lose weight, and for diet supplements that can prevent weight gain and maintain a healthy body weight.

Capsaicin is the main ingredient in chili peppers, responsible for the characteristic “hot” sensation that this spice evokes in the human mouth [4]. Capsaicin is eaten on a daily basis by an estimated quarter of the world‘s population. This is surprising since the same “hot” taste which is found pleasurable by many humans repels most animals [4]. It was posited that the pepper pod uses capsaicin as a chemical weapon to deter herbivores [5]. Indeed, capsaicin is added to bird-feed to keep it safe from squirrels and other rodents [6].

Many theories were formulated to explain the popularity of culinary capsaicin. One theory speculates that capsaicin may help keep the body lean [7]. This theory is probably inspired the popular articles that promote capsaicin as a “magic bullet” for weight loss [8]. This is an attractive theory since capsaicin has been a culinary staple for thousands of years and is generally deemed safe when it is consumed in hedonically acceptable, restaurant-like doses. But is this theory true? This review aims to critically evaluate the available evidence for and against capsaicin as a weight control agent.

## 2. Dietary Capsaicin in Animal Experiments

A large number of studies have examined the effects of capsaicin on appetite control and weight gain. These studies can be divided into three major groups: (1*) per os* capsaicin in healthy rodents that are kept on normal chow, (2) capsaicin that is added to high fat diet (HFD), and (3) dietary capsaicin in animal models of diabetes and obesity.

In early subchronic toxicity studies, pure capsaicin (50 mg/kg per day) or capsicum crude fruit extract (0.5 g/kg day) that was given to rats via stomach tube for 60 days had no effect on food intake but reduced the weight gain of the animals [9]. This was attributed to decreased fat absorption from the gastro-intestinal (GI) tract [10]. In accord, after one month of capsaicin administration, significant reductions in serum glucose, triglyceride, and cholesterol levels were observed in these animals [9]. Interestingly, after two weeks habituation to capsaicin (250 μg/g chow), the rats preferred the “hot” chow to plain food [11]. This observation implies that the “hotness” of capsaicin is an acquired taste in men.

In a 12-week study with mice kept on normal chow or HFD (32% animal lard, corresponding to 60% energy from fat), capsaicin (2 mg/kg/day, *per os*) reduced weight gain by approximately 50% in the HDF group (Figure 1A,B), but had no effect on the body weight in animals that consumed normal chow (Figure 1A) [12]. Furthermore in rats, dietary capsaicin (0.014–0.028%) suppressed visceral (for example, perirenal) fat accumulation in a dose-dependent manner [13]. Dietary capsaicin also lowered the serum triglyceride levels both in rats [13,14] and guinea pigs [15] on HFD. Importantly, the latter action was accompanied by an up to 70% decrease in atherosclerotic plaque formation [15].

Some of these beneficial effects were recapitulated by dietary capsiate (10 mg/kg) [16] and evodiamine (3 mg/kg) [17], two non-pungent capsaicin analogues.

In mice that were fed HFD for 12 weeks, capsiate (2 or 10 mg/kg) blocked weight gain by reducing lipid accumulation in the white adipose tissue [18]. In other studies, however, capsiate had no effect on weight gain of the animals that were fed a HFD ad libitum unless combined with exercise training [19].

In male C3H mice, the evodiamine-induced visceral fat loss was accompanied by serum free acid and liver triglyceride levels that were lower than those that were measured in the control group [17]. Evodiamine is an alkaloid that is found in *Evodia rutaecarpa*, a fruit that is used in traditional Chinese medicine for digestive problems. Evodiame activates the capsaicin receptor Transient Receptor Potential, vanilloid-1 (TRPV1), although it lacks the characteristic “hot” taste of capsaicin [20].

In C57BL/6J mice, capsaicin (0.4 mg/kg) that was fed in combination with menthol (20 mg/kg) and cinnamaldehyde (2 mg/kg) also prevented HFD-induced weight gain [21]. This dietary protocol is known as the “triagonist regimen” since capsaicin is the archetypal agonist of TRPV1 [22], menthol activates TRPM8 [23,24], and cinnamaldehyde acts via TRPA1 [25]. When combined with eicosapentaenoic acid (an omega-3 fatty acid that is found in cold-water fish), capsaicin (0.01%) reduced weight gain both in the HFD (32% fat) and standard chow (5.3% fat) groups [26].

In the NAFLD mouse (a murine model of non-alcoholic fatty liver disease [27]), dietary capsaicin suppressed liver fat accumulation [28]. Furthermore, in congenic, spontaneously obese, and diabetic KKAγ mice [29], capsaicin (0.015%) that was added to HFD for 3 weeks reduced fasting glucose, increased adiponectin, and suppressed weight gain [30]. However, in the genetically obese ob/ob mice [31], capsaicin diet (0.02% for 6 weeks) had no measurable effect on weight gain, despite the improved glucose homeostasis [32].

So how do these capsaicin doses compare to dietary capsaicin consumption in men? An average female laboratory rat weighs approximately 250 g. Thus, a 50 mg/kg body weight dose translates into 12.5 mg per day. This is somewhat higher than the average daily dietary capsaicin consumption in Korea (0.6 to 3 mg), although some Koreans consume as much as 20 mg capsaicin per day [33]. Moreover, the 0.01% capsaicin content that is used in rat chow is comparable to the diet of rural Thai people [34].

In conclusion, dietary capsaicin in doses that are comparable to human consumption may reduce body fat accumulation in rodents on HFD, but not on normal chow (Figure 1). It is not clear, however, if animals that are already obese can lose weight if they are placed on a capsaicin diet.

## 3. Dietary Capsaicin in Human Studies

In a cross-over study with 15 young, non-obese volunteers (average age, 29.7 years; body mass index [BMI] = 23.3 kg/m^2^), dietary capsaicin (2.56 mg with every meal, corresponding to 39 thousand Scoville units) increased fullness and depressed the desire to eat, resulting in a 25% reduction in the energy balance [35]. In a second study with 24 subjects with a normal body weight (12 males and 12 females; average BMI = 25 kg/m^2^), 0.25% capsaicin (80 thousand Scoville units) in either tomato juice or capsule increased satiety and diminished the energy intake from 11.5 mega joules (MJ)/day (placebo group) to 9.9 MJ/day (capsaicin group) [36]. The capsaicin effect on satiety was attributed to GI distress [37].

Unlike in mice, this beneficial capsaicin effect was not mimicked by capsiate. In a one-month study with 78 healthy volunteers, 3 or 9 mg dihydrocapsiate showed only minimal thermogenic action (50 kcal/day) which was in the range of normal day-to-day variability [38]. A meta-analysis of seven clinical studies confirmed the marginal effect of dietary capsiate on weight gain, with uncertain long-term sustainability [39].

Interestingly, in 20 young men, dietary capsiate (12 mg per day) increased upper body strength by 13.4% over a 6-week period, a significant increase (*p* = 0.041) over the placebo effect (5.6%) [40]. This capsiate effect was, however, accompanied by a modest, on average 1 kg, increase in body weight [40].

In young obese individuals (BMI > 41.5 kg/m^2^), 2 mg capsaicin had no hypophagic effect (that is, no noticeable change in hunger or satiety), although it increased the post-meal resting energy expenditure from 1957 kcal/day to 2342 kcal/day as measured by computerized calorimetry [41]. No change was noted in serum ghrelin, peptide YY, and glucagon-like peptide levels [41].

A meta-analysis of the various clinical studies yielded similar conclusions. In a meta-analysis of nine clinical studies, capsaicin was found to increase energy expenditure by 70 kcal/day in men with a BMI > 25 kg/m^2^, but not in those with a BMI < 25 kg/m^2^ [42]. Another meta-analysis of 10 studies with 191 participants found decreased calorie intake that was attributed to altered food preference from fat to carbohydrate [43]. A meta-analysis of 13 studies that was performed between 1990 and 2019 concluded that both “hot” capsaicin and “mild” capsinoid can increase the resting metabolic rate and energy expenditure due to a rise in fat oxidation, although the reported changes were very modest (Table 1) [44]. Importantly, this study also found bias (selection, performance, detection, attrition, and/or reporting) in all of the clinical studies that were analyzed [44].

Other studies combined capsaicin with other putative weight control agents. For example, in the METABO study (dietary capsaicin combined with raspberry ketone, caffeine, garlic, ginger, and citrus auratium), an 8-week clinical trial with 45 obese, but otherwise healthy individuals, only a 2% reduction in weight gain compared to the placebo was detected [51]. A second study involving 80 obese individuals with a BMI > 31 kg/m^2^ reported a 4% loss in body weight on a diet containing capsaicin, caffeine, tyrosine, and catechines [52]. It is not clear to what degree capsaicin might have contributed to these modest weight losses.

One should note that most of these studies used a capsaicin dose of 2 mg/meal or higher. A meta-analysis even emphasized that capsaicin effects only occur with a minimal dose of 2 mg per meal [43]. Indeed, CAPSIMAX (a product of OmniActive Health Tech.) contains 2 mg capsaicin per tablet. This dose (4 mg per day, assuming two “hot” meals per day) appears to be a bit high by Western standards since the estimated maximum daily capsaicin intake by mild chili peppers in Europe is 1.5 mg/day.

In a clinical trial with 77 young (on average, 30 y.o.) overweight individuals (BMI, 27.5 kg/m^2^) who took CAPSIMAX, one or two tablets a day (2 or 4 mg) for 12 weeks, the self-reported energy intake decreased by 257 kcal/day, resulting in a more favorable waist to hip ratio by the end of the study [53].

There were two large epidemiologic studies that examined the BMI of people who like “hot” cuisine and those who prefer their meals to be non-spicy, and neither found any significant difference. The American study [54] was based on the National Health and Nutritional Examination Survey, in which 16,179 participants were followed for 6 years (1988 to 1994). This study found an obesity rate of 23.9% among chili eaters (4107 individuals) as opposed to 25.4% among non-eaters (12,071 individuals). Nonetheless, the mortality rate among chili-eaters (21.6%) was clearly superior to that (33.6%) of the non-eaters. The Chinese study [55] enrolled close to half-million people and reported BMI values (kg/m^2^) of 23.2, 23.5, 23.6, and 23.6, respectively, in groups that consumed less than 1, 1 to 2, 3 to 5, or 6 to 7 weekly spice meals. Interestingly, this study also reported a health benefit for chili eaters, for example, lower cancer and ischemic heart disease rates.

In healthy volunteers, capsicum (5 g per day) decreased plasma glucose levels compared to a placebo [56]. Preclinical studies in diabetic animals suggested that dietary capsaicin may improve glycemic control [30,32], implying a therapeutic potential in patients with T2DM. Indeed, traditional healers use hot pepper for this purpose [57]. Unfortunately, a recent meta-analysis of 14 controlled clinical trials with capsaicin supplementation found neither beneficial nor detrimental effects on blood glucose and insulin levels [58].

In conclusion, capsaicin effects on appetite are modest and their long-term influence on body weight is questionable. Indeed, no significant difference in BMI was found between chili-eaters and non-eaters [54,55].

## 4. Dietary Capsaicin: Mechanisms of Action

There are four major models that have been promoted to explain the beneficial effects of culinary capsaicin on weight gain (Figure 2). According to the first and oldest theory, capsaicin exacerbates intestinal passage and thereby reduces the absorption of calories [59]. This model is in keeping with the well-known ability of hot, spicy food to cause diarrhea in infants [60] and sensitive individuals [61]. The second theory posits that capsaicin can boost thermogenesis [62,63]. The third model connects capsaicin-sensitive visceral afferents to the arcuate nucleus [64], the center of appetite regulation. Finally, the fourth theory that has gained popularity recently assumes that capsaicin can change the gut microbiota in a way that may help maintain a healthy body weight [65]. Since the effect of capsaicin on gut bacteria is not mediated by TRPV1 (so in a way it is non-specific), it will be discussed below under a separate subheading.

The GI tract is densely innervated by capsaicin-sensitive (TRPV1-expressing) nerves that sense visceral pain (afferent function) and regulate intestinal motility (efferent function) [67,68]. In rats, dietary capsaicin stimulates mucus production in the colon that, in turn, may reduce fat absorption [12]. In men, capsaicin hastens the intestinal transit of the meal [69,70], although it has no effect on gastric emptying [71]. This is in accord with the reduced fat absorption that was noted in rats when capsaicin was added to the chow [10]. Furthermore, capsaicin can cause gastrointestinal distress that, in turn, may suppress the desire to eat more [37,72]. Somewhat unexpectedly, epithelial cells in the human gut were also shown to express functional TRPV1 [73,74,75], implying a direct capsaicin effect on the GI mucosa.

There is good evidence that capsaicin can boost thermogenesis, both shivering and non-shivering [62,63]. In experimental animals, capsaicin can induce hypothermia and initiate counter regulatory mechanisms, such as shivering, to generate heat [76,77]. In men, dietary capsaicin is known to induce “gustatory sweating” [78]. To explain the popularity of “hot” food under tropical climates, it was speculated that capsaicin can cool the body (capsaicin as “natural air conditioner”) [79].

Of note, visceral TRPV1-expressing afferents have been implicated in the thermoregulatory action of capsaicin [80]. Thus, the activation by capsaicin of the same afferents may reduce the calorie intake by speeding the intestinal transport of food, and, at the same time, increase energy expenditure by lowering the body temperature. Furthermore, these vagal afferents when stimulated by capsaicin may activate appetite-suppressant CART (cocaine, amphetamine, and proopiomelanocortin-regulated) neurons in the arcuate nucleus [81]. It should be noted here that CART neurons themselves express TRPV1, therefore, they may also be directly activated by capsaicin [82].

Brown adipose tissue (BAT) is a key player in non-shivering thermogenesis [83,84]. Indeed, there is an inverse relationship between BAT and obesity. Dietary capsaicin was reported to “brown” the white adipose tissue [62,85,86]. For example, near-infrared time-resolved spectroscopy (NITR) detected a 46% increase in brown adipose tissue in 20 volunteers on daily capsaicin for 8 weeks [87].

Both murine and human visceral adipose tissue seem to express TRPV1 [88]. In mice, capsaicin increased the phosphorylation of sirtuin-1 (SIRT-1) [89], a protein that is involved in lipid metabolism [90]. This effect was prevented by capsazepine, a TRPV1 antagonist [91], and was absent in TRPV1-null mice [89], indicating a TRPV1-mediated capsaicin action. Furthermore, capsaicin protected the expression of the thermogenic genes, *ucp-1*, *bmp8b*, *pgc-1α*, and *prdm-16*, from HFD-induced downregulation [89]. In a second study, TRPV1 activation elevated UCP-1 (uncoupling protein-1) protein content in the brown adipose tissue of the mouse and protected the animals from HFD-induced visceral fat accumulation [92]. Since UCP-1 is a known player in the “browning” of the white adipose tissue [93], this observation may provide a mechanistic explanation for fat “browning” by capsaicin [86].

This is interesting animal research, but is it relevant for dietary capsaicin actions in men? In other words, can dietary capsaicin reach human adipose tissue in concentrations that are high enough to replicate the effects that are seen in mice? Probably not. Although capsaicin is readily absorbed from the GI tract of both rats [94] and men [95,96], it is rapidly metabolized in the liver, producing 126 transformation products [97,98]. Although some of the capsaicin metabolites were detected in human urine [98], it is not known if any of these compounds may mimic capsaicin actions on adipose cells. In human volunteers, the half-life of capsaicin in the blood was approximately 25 min with a peak concentration of 8.2 nM [96]: this should be compared to the affinity of capsaicin for human TRPV1, 640 nM [99].

## 5. Capsaicin and Gut Microbiota

The effect of the gut microbiota on health and disease is subject to intense research. Gut bacteria play a critical role in colonic health [100]. What we eat will either help maintain a healthy microbiota in the colon, or cause disease by killing “good” bacteria and supporting the growth of pathogens. There is a growing body of evidence linking abnormal gut microbiota [101] and a malfunctioning gut-microbiota-brain axis [102] to obesity. Indeed, a transplant of fecal microbiota from adult twins discordant for obesity to germ-free mice recapitulated the phenotype (lean versus obese) of the fecal donor [103]. Importantly, the mice could be rescued from fecal transplant-induced weight gain by microbiota of the lean animals [103]. These findings reveal transmissible and modifiable effects of the gut microbiota on obesity.

Body fat content seems to correlate with gut microbiota diversity: the higher the fat, the lower the diversity [104]. For instance, obese people host more firmicutes and fewer *Akkermansia* in their GI tract [105]. Conversely, weight loss interventions increase the abundance of *Akkermansia*, an “antiobesity” bacterium, in the stool [106]. Many pre- and probiotics that are used to treat obesity were shown to restore the normal diversity of the gut microbiome [107]. Capsaicin may correct this “dysbacterosis” via multiple mechanisms as an alternative to “fecal transplant” [108].

In the C57BL/6J mouse, capsaicin (2 mg *per os*) was reported to increase the contribution of *Akkermansia* to the gut microbiota, presumably by stimulating mucin production in the colon [12]. In this context, it may be relevant that mucin-producing columnar epithelium in the colon expresses TRPV1 [109]. However, this capsaicin effect was seen both in the TRPV1-null and wild-type animals, indicating a non-specific (that is, not capsaicin receptor-mediated) action [108]. Capsaicin may also improve the bacterial diversity by eliminating pathogens directly [110].

Chronic low-grade inflammation has been implicated in the pathomechanism of obesity and metabolic syndrome [111]. The dysbacteriosis, in particular the overgrowth of the S24-7 bacterium family that occurs in the obese, can lead to metabolic endotoxemia, which, in turn, may maintain this low-grade inflammation [112]. Capsaicin may represent a novel dietary strategy to prevent endotoxemia [113]. Indeed, dietary capsaicin was reported to reduce the number of Gram-negative lipopolysaccharide (LPS)-producing bacteria in the stool [113]. Importantly, in this study the capsaicin-fed animals gained less weight when they were kept of HFD, and the protective effect of capsaicin was transferable to mice on a control diet by fecal transplant [113].

Last, LPS has been shown to both directly activate TRPV1 [114] and to indirectly sensitize capsaicin-sensitive afferents [115], creating a “microbe-gut-nerve-brain axis”. This loop may be disconnected capsaicin-desensitization.

In conclusion, dietary capsaicin may aid in restoring the “pro-lean” gut microbiota.

## 6. Dermal Capsaicin Patch for Weight Loss?

The topical application of 0.075% capsaicin cream to the skin of mice that were fed HFD significantly reduced weight gain and visceral fat [116]. Topical capsaicin also reduced serum glucose and triglyceride levels [116]. Furthermore, ovalbumin-allergic mice that were treated topically with a 0.075% capsaicin cream displayed attenuated food allergy symptoms (e.g., reduced blood eosinophilia and IgE levels), restoring normal appetite and body weight [117].

Topical capsaicin may be an attractive approach for people who either dislike “hot” spicy food or experience GI symptoms such as abdominal pain and bloating after eating it. However, this observation with low concentration (0.075%) capsaicin is difficult to apply to men since the analgesic capsaicin patch employs a 100-fold higher (8%) concentration.

Although topical capsaicin is readily absorbed from the human skin [118], its concentration in the blood probably stays low. Indeed, the horse-sensitive methods (e.g., UHPLC and MS) failed to detect any capsaicin in the blood after treating the skin of the animals with 0.1% capsaicin for 5 days [119]. Furthermore, dermal capsaicin evokes a blunted pain and flare response in the skin of obese individuals, indicative of reduced capsaicin sensitivity [120]. High-dose (8%) capsaicin patches are in clinical use to relieve chronic neuropathic pain [121]. As yet, no clinical study with dermal capsaicin creams or patches have been done to test any effect on body weight.

## 7. Capsaicin-Sensitive Nerves and Obesity

Capsaicin-sensitive neurons are bi-directional neurons with somata in sensory (dorsal root and trigeminal) ganglia [4,67,68]. The peripheral terminals of these neurons are sites of release for neuropeptides (for example, substance P, SP, and calcitonin gene-related peptide, CGRP) that initiate the biochemical cascade that is known as neurogenic inflammation [4,68]. The central efferents enter the spinal cord where they make synapse with second order neurons of the dorsal horn [4,68]. These efferents convey nociceptive information into the central nervous system. The role of these neurons in pain sensation, and the efforts to develop clinically useful analgesic agents that block the capsaicin receptor TRPV1, were detailed elsewhere [122,123,124]. Here it suffices to mention that the initial excitation by capsaicin of these neurons is followed by a lasting refractory state, traditionally termed “desensitization” [4,68]. Neonatal capsaicin administration can also kill these neurons [125]. As a tool to dissect the function of capsaicin-sensitive afferents, desensitization of adult animals seems to be preferable since rodents that grow up without such nerves due to neonatal treatment may develop compensatory mechanisms. Indeed, newborn rats whose TRPV1-expressing neurons had been eliminated by capsaicin (50 mg/kg s.c.) show no change in body weight as adults compared to solvent controls [126]. This is in sharp contrast to rats that were desensitized to capsaicin as adults: these animals stay lean because they resist aging-associated weight gain [127]. Gaining weight in the elderly has been linked to increasing circulating CGRP levels [128]. Capsaicin desensitization depletes CGRP [4,68], and thus may prevent the age-related increase in circulating CGRP.

With intraperitoneal capsaicin administration (5 mg/kg), the visceral vagal afferents can be selectively desensitized with no global effect on capsaicin-sensitive neurons. After this intervention, rats were deprived of food for 5 days. During these 5 days of food deprivation, the capsaicin-treated animals lost 18.9 g whereas the controls lost 15.8 g: that is, the capsaicin group showed 20% more weight loss [129].

Visceral fat is thought to be more deleterious than subcutaneous adipose tissue. Intact capsaicin-sensitive intestinal afferent function seems to be essential for the redistribution of fat from viscera to subcutis [130].

## 8. The Confusing Phenotype of the Trpv1-Null Mouse

Unlike wild-type mice that avoid capsaicin-flavored food [6], TRPV1-null animals readily consume hot habanero pepper (Figure 3C). On low-fat (4.5%) diet, no difference in weight gain was observed between the TRPV1-null and the wild-type mice [131]. By contrast, on HFD (11% fat), TRPV1-null mice gained less weight (34 g) than the wild-type controls (51 g) [131]. Importantly, the TRPV1-null animals had less visceral fat and also showed reduced fatty change in the liver [131]. This study concluded that a lack of TRPV1 protects against diet-induced obesity.

However, a second study [132] came to a very different conclusion. Young TRPV1-null mice stayed lean and physically active (Figure 3A) but becoming old these animals became obese (Figure 3B) and “lazy”, with enhanced hepatic steatosis [132,133]. The aging TRPV1-null mice even gained more weight on normal chow (Figure 4). Adding to the confusion, a third study found no difference in weight gain between TRPV1-null and wild-type mice [134].

Three studies, three very different outcomes. The genetic background of TRPV1-null mice may be a factor in the discrepant results, along with the age of the animals and the dietary differences.

Moreover, it is not easy to reconcile the genetic *Trpv1* inactivation studies with the capsaicin desensitization experiments: the first exacerbates aging-associated obesity whereas the second seems to protect against it. One should keep in mind that the TRPV1-null mice lack the capsaicin receptor but have otherwise functional capsaicin-sensitive neurons. By contrast, in the capsaicin-desensitized animals these nerves are non-functioning.

## 9. Obesity-Related Changes in Trpv1 Expression

As mentioned above, non-neuronal TRPV1 is expressed in fat cells [86]. When rats were kept on HFD, diminished TRPV1 expression was noted both in the white and brown adipose tissue. Human visceral adipose tissue also possesses TRPV1, with markedly reduced expression in the obese [86]. Morbidly obese men also showed diminished pain and flare response in the capsaicin skin test [120], though it is not completely clear if this reflects reduced TRPV1 expression or obesity-induced small sensory fiber neuropathy.

TRPV1-like immunoreactivity was described in the human GI tract, from the stomach (Figure 5) through the duodenum to the colon [73,74,75]. In epithelial cells, no difference in TRPV1 expression was found between obese (BMI > 40 kg/m^2^) and control individuals [75].

## 10. Conclusions

Based on the available literature, one may conclude that spicy food (culinary capsaicin), though it may suppress appetite in the short run, does not protect against obesity in the long run. Indeed, two large epidemiological studies found no significant difference in BMI between chili eaters and non-eaters [54,55]. This is hardly unexpected since capsaicin effects are known to undergo desensitization upon repeated challenge [4,68]. It also implies that putative TRPV1 agonists that do not induce desensitization may prove clinically useful in appetite control.

Furthermore, dietary capsaicin does not appear to be an effective weight loss agent in the obese. This may in part be related to the reduced TRPV1 expression in the visceral adipose tissue of fat people [87].

Dietary capsaicin reduced blood glucose, cholesterol, and triglyceride levels in experimental animals [14,15,30,32], but not in clinical studies [58].

That said, dietary capsaicin is not without health benefits. Indeed, chili-eaters were found to live longer and stay healthier than their compatriots who favored bland food [54,55].

In conclusion, dietary capsaicin is not a magic “exercise in a pill”, though there is good evidence that capsaicin-sensitive afferents play an important role in body weight regulation. The challenge is to manipulate these nerves for weight control without causing unacceptable side-effects.

## 11. Future Directions

For future research, there are several intriguing observations that may be worth pursuing. First, experiments with laboratory animals suggest that dietary capsaicin can reduce weight gain in the young and lean on HFD, but not in older individuals on HFD, in the obese, or those (regardless of age or body weight) who are on a regular diet. Future clinical studies will test the human relevance of these observations. If this model holds true in men, capsaicin may help prevent childhood and juvenile obesity.

Second, capsaicin seems to have a beneficial direct effect on gut microbiota, by promoting the growth of “anti-obesity” bacteria and eliminating undesired pathogens [108,109,110,113]. Here the challenge is to deliver sufficiently high capsaicin doses without causing abdominal discomfort or diarrhea. This will be no longer a concern if the capsaicin effects on the gut microbiota can be replicated by non-pungent capsaicin congeners such as capsiates.

Third, capsaicin-desensitized animals are protected from aging-associated weight gain [127]. Chronic, low-grade inflammation has been implicated in the pathomechanism of obesity [135], and capsaicin-sensitive nerves may play a key role in maintaining this inflammation by releasing pro-inflammatory neuropeptides such as CGRP [4,68]. If so, molecules that block the activation of capsaicin-sensitive neurons may ameliorate aging-associated obesity.

Fourth, small molecule TRPV1 antagonists are potential analgesic [122,123,124] and antidiabetic [111] drugs. Some of these compounds have already entered Phase-3 clinical trials. What effect, if any, can such drugs have on body weight, especially in the elderly?

Last, the topical application of capsaicin cream to the skin of mice that were fed HFD significantly reduced weight gain and visceral fat accumulation, and normalized the serum glucose and triglyceride levels [116]. Since high-dose capsaicin dermal patches are already available for pain control [121], it would be interesting to see if these patches have a similar beneficial action in men.

## Figures and Tables

**Figure 1 pharmaceuticals-15-00851-f001:**
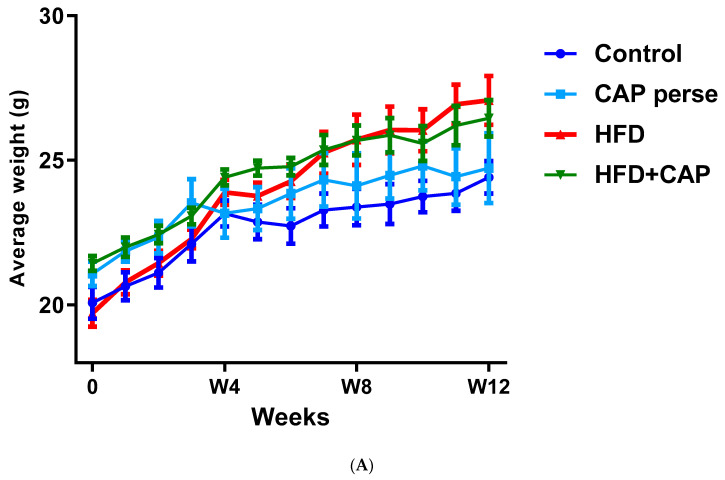
In C57BL/6 mice, capsaicin (2 mg/kg/day, *per os*) reduced weight gain by approximately 50% in animals that were kept on HFD (**A**,**B**), but had no effect on body weight in the normal chow group (**B**). Figure courtesy of Dr. Mahendra Bishnoi (based on [12]). ** *p* < 0.01 versus control animals, ## *p* < 0.01 versus HFD animals.

**Figure 2 pharmaceuticals-15-00851-f002:**
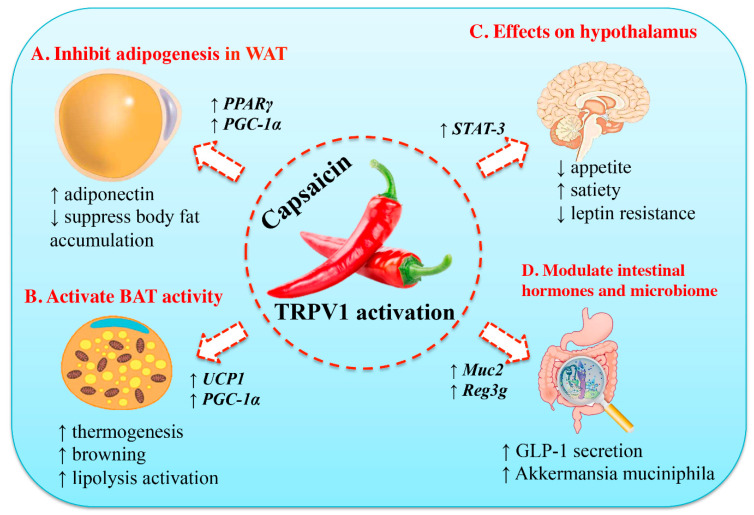
Molecular mechanisms of the anti-obesity effects of capsaicin. (**A**) Capsaicin can inhibit adipogenesis in preadipocyte and adipocyte by up-regulating the expression of peroxisome proliferator-activated receptor gamma (PPARγ) and uncoupling protein-1 (UCP-1). Thus, it will stimulate adiponectin secretion and increase body fat accumulation. (**B**) Capsaicin can activate brown adipose tissue (BAT) activity, accompanied by increased expression of UCP-1 and PPAR-coactivator-1α (PGC-1α). (**C**) Capsaicin can suppress appetite, increase satiety, and ameliorate insulin resistance. (**D**) Capsaicin can modulate its function in the gastrointestinal tract and gut microbiome, including stimulation of glucagon-like peptide-1 (GLP-1) secretion and increase in population of the gut bacterium *Akkermansia muciniphila*. STAT-3, and signal transducer and activator of transcription-3 (STAT3). Reproduced with permission from [66].

**Figure 3 pharmaceuticals-15-00851-f003:**
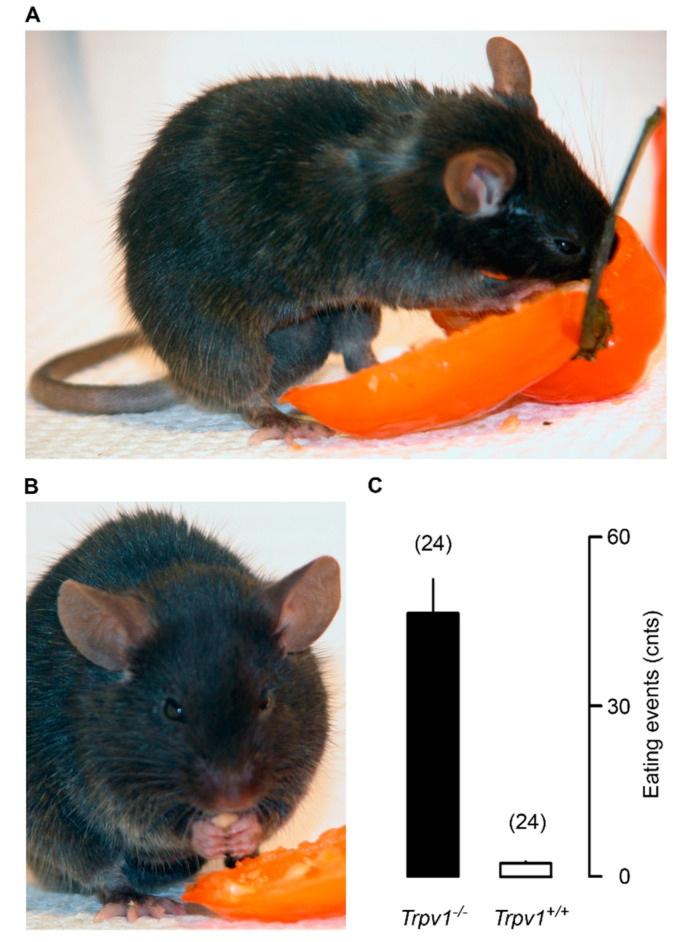
*Trpv1* knockout mice are insensitive to capsaicin and eagerly consume habanero chili pepper (panel **A**), including the hottest parts such as the septa and the seeds (panel **B**). Panel (**C**) shows the mean number of pepper-eating events during a 20-min period. Reproduced with permission from [132].

**Figure 4 pharmaceuticals-15-00851-f004:**
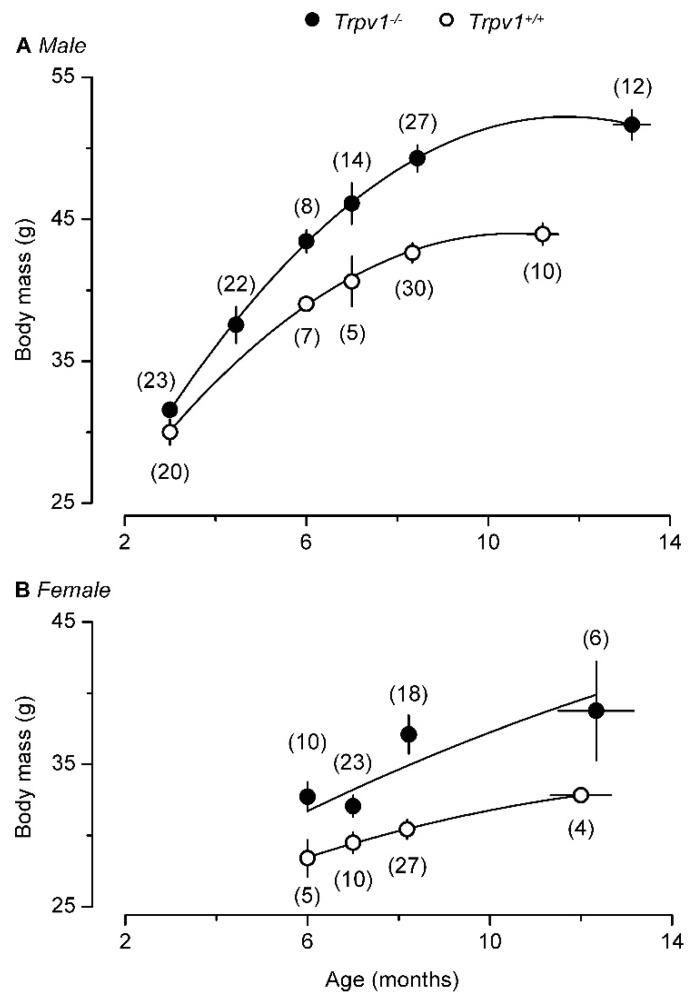
With age, both male (**A**) and female (**B**) TRPV1-null mice become heavier than their age-matched, wild-type controls. Reproduced with permission from [132].

**Figure 5 pharmaceuticals-15-00851-f005:**
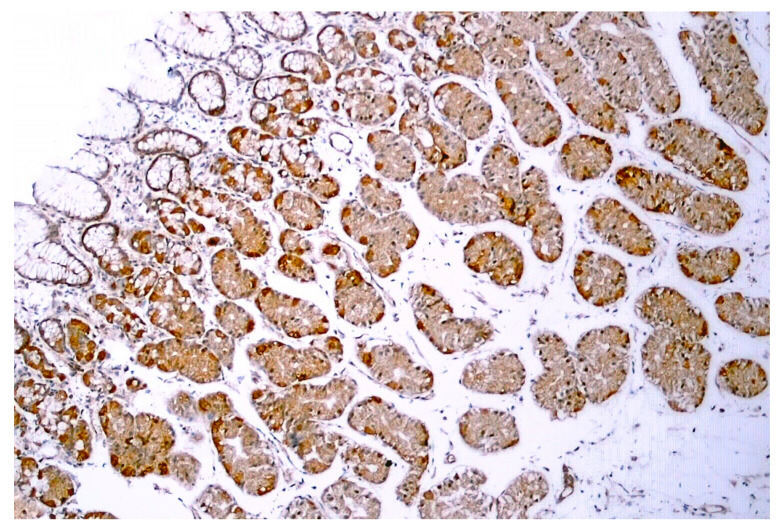
TRPV1-like immunoreactivity in human gastric biopsy (normal body weight individual). Figure courtesy of Dr. Gülsüm Özlem Elpek, Akdeniz University, Turkey.

**Table 1 pharmaceuticals-15-00851-t001:** Weighted mean difference in the resting metabolic rate (kcal/day), energy expenditure (kcal/day), fat oxidation (g/hour), and carbohydrate oxidation (g/hour), in seven clinical studies with dietary hot pepper (capsaicin) [45,46,47] or non-pungent capsinoid [38,48,49,50]. These studies reported a modest but consistent increase in the resting metabolic rate that may be attributed to a rise in fat oxidation, with no difference between hot pepper and mild pepper consumption.

Reference	Resting Metabolic Rate (kcal)	Energy Expenditure	Fat Oxidation	Carbohydrate Oxidation
[45]	24.09	3.06	0.06	−1.05
[46]	34.41	6.96	0.12	−3.04
[47]	23.87	23.9	1.00	N.D.
[48]	47.51	N.D.	0.61	4.33
[49]	8.50	14.38	0.54	N.D.
[38]	43.00	N.D.	0.29	N.D.
[50]	53.60	N.D.	0.05	N.D.

## Data Availability

Data sharing not applicable.

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
