# Peer review of "Capsaicin for Weight Control: “Exercise in a Pill” (or Just Another Fad)?"

_pharmaceuticals, 2022, doi:10.3390/ph15070851_

Round 1

Reviewer 1 Report

Dr. Arpad Szallasi presents a review on, the manuscript is very good, it is well structured and addresses the role that capsaicin has on weight control both for and against and supports his arguments in what has recently been published in the literature, I think which is suitable for publication. I only have a couple of observations that I list below

Introduction section, Parr 3 please add: common diseases like hypertension, hyperinsulinemia, insulin resistance and T2DM, pathologies that comprise the metabolic syndrome (SM).

Dietary capsaicin in human studies section. Please define “MJ” and “BMI”

Dietary capsaicin mechanisms of action section: at the beginning of the manuscript it was already abbreviated the gastro-intestinal (GI), please in the parr 2 delete gastro-intestinal and leave the abbreviation. In same section parr 5 please change the order first is near-infrared time-resolved spectroscopy and then the abbreviation

Capsaicin sensitive nerves and obesity section. Please define CNS

The manuscript presents four well-represented figures. However, based on the information presented, I believe that the information would be reinforced if a couple of tables were added that summarized the most important sections in the manuscript.

Reviewer 2 Report

In this article, the authors made a comprehensive review on the association with capsaicin and weight control in basic and clinical aspects. Overall, the review is informative and interesting. I have a few suggestions. The authors focused on the weight reduction and adipose tissue mass altered by capsaicin ingestion. It is also important to include the information on the effects of capsaicin on serum lipid profiles and glycemic status described in the literatures. With regard to the mechanisms of capsaicin, epigenetic regulation, e.g. Sirt1, was mentioned in the text, should be included in Figure 1. In Figure 3, The authors stated that “the TRPV1 null animals had less visceral fat and also showed reduced fatty change in the liver [119]. This study concluded that lack of TRPV1 protects against diet-induced obesity”. But, both male and female TRPV1 null mice showed more body mass that wild-type mice. Clarification is needed. In Figure 4, was the IHC of gastric biopsy from an overweight subject? It is better to include normal and high BMI subjects.

Reviewer 3 Report

The authors critically analyzed the effect of Capsaicin in weight control.

I would like to start by mentioning that a word template is available on the journal website. I suggest the authors to read carefully the instructions provided before to submit the manuscript.

However, the authors of the study review the papers in a critical manner. In a general key, the available data are presented in a critical manner and all the major point were touched.

The conclusion section must be rephrased/reorganized in full. It looks more like a discussion section. Moreover, the question is redundant. I prefer to understand what is the authors opinion after reviewing the literature instead of putting questions.

If the manuscript is reorganized according with the journal template, and the conclusion section is rephrased I suggest the editors to continue the review process.

Round 2

Reviewer 3 Report

I have no comments. I found the manuscript may be considered for being published.

Author Response

thank you